# Methionine in a protein hydrophobic core drives tight interactions required for assembly of spider silk

Julia C. Heiby [1,5], Benedikt Goretzki [2,3,5], Christopher M. Johnson [4], Ute A. Hellmich [2,3]* & Hannes Neuweiler [1]*

Web spiders connect silk proteins, so-called spidroins, into fibers of extraordinary toughness. The spidroin N-terminal domain (NTD) plays a pivotal role in this process: it polymerizes spidroins through a complex mechanism of dimerization. Here we analyze sequences of spidroin NTDs and find an unusually high content of the amino acid methionine. We simultaneously mutate all methionines present in the hydrophobic core of a spidroin NTD from a nursery web spider's dragline silk to leucine. The mutated NTD is strongly stabilized and folds at the theoretical speed limit. The structure of the mutant is preserved, yet its ability to dimerize is substantially impaired. We find that side chains of core methionines serve to mobilize the fold, which can thereby access various conformations and adapt the association interface for tight binding. Methionine in a hydrophobic core equips a protein with the capacity to dynamically change shape and thus to optimize its function.

--------------------------------------------------------------------------------------------------------------------------

[1] Department of Biotechnology and Biophysics, Julius-Maximilians-University Würzburg, Am Hubland, 97074 Würzburg, Germany. [2] Institute for Pharmacy and Biochemistry, Johannes-Gutenberg-University Mainz, Johann-Joachim Becherweg 30, 55128 Mainz, Germany. [3] Center for Biomolecular Magnetic Resonance, Goethe-University, Max-von-Laue-Strasse 9, 60438 Frankfurt, Germany. [4] Medical Research Council Laboratory of Molecular Biology, Francis Crick Avenue, Cambridge CB2 0QH, UK. [5] These authors contributed equally: Julia C. Heiby, Benedikt Goretzki. *email: u.hellmich@uni-mainz.de; hannes.neuweiler@uni-wuerzburg.de

Over hundreds of millions of years spiders evolved the construction of silk webs of different geometries tailored for various purposes including prey capture, reproduction and shelter[1]. They use up to seven glands specialized for these purposes. In each of them soluble silk proteins, so-called spidroins, are being tightly connected during their passage through the spinning duct and brought out of solution in a controlled fashion to form a fiber[2]. Spiders evolved mechanisms that carefully control the ordered phase and structural transitions in order to assure fabrication of high-quality material and to avoid early and lethal fibrillation within the gland. Dragline silk, which is used as a lifeline and to build the web frame, is formed by spidroins from the major ampullate gland (MaSp). The dragline represents the toughest known biological thread and is thus a current focus of biomimetic material sciences[3,4].

To date, the molecular mechanisms that underlie phase and structural transitions of spidroins are only understood in parts. At the beginning of the process, spidroins are stored in soluble form in the ampulla of a spinning gland, which is located in the spider's abdomen. On demand, they pass through the tapering duct where they experience mechanical and chemical stimuli that transform them into silk[2–4]. The N-terminal and C-terminal domains of spidroins (NTD and CTD) fulfill critical functions during storage and assembly[5]. Both domains are highly conserved five-helix bundles that provide water-solubility and connectivity[5]. The CTD covalently connects two spidroins through formation of a homo-dimer[6,7]. The NTD, on the other hand, contains a relay that triggers dimerization upon a change of solution conditions within the spinning duct[4,8]. A decrease of pH along the duct leads to tight self-association of the NTDs, which connects and poly-merizes spidroins[9–12]. The mechanism of NTD dimerization is conserved across glands and species and involves site-specific protonation of surface charges and conformational change[4,8–17]. While the contribution of surface charges and their protonation to dimerization has been investigated intensely[9,10,12,13,16], the mechanism of conformational change and its role in dimerization remains largely unexplored. Conformational change involves motion of helices that are part of the association interface, which tilt upon dimerization and adopt perfectly self-complementary surfaces[14,15]. Helix rearrangement is associated with motion of a single, conserved tryptophan (Trp) from buried to solvent-exposed position[4,14,17]. The driving force underlying these con-formational changes and their energetic contribution to spidroin association, and hence spider silk formation, are unknown.

Here we find that side chains of the amino acid methionine (Met), which are present in unusually high numbers in the core of the NTD, are responsible for conformational changes of the domain. We simultaneously mutate all core Met in the NTD of MaSp1 from the nursery web spider *Euprosthenops australis* to leucine (Leu) and make surprising observations. The mutant gained a substantial amount of stability compared to the wild-type protein and its structure is fully preserved. Its ability to dimerize is considerably impaired. Conformational dynamics of the mutant are stalled, in contrast to the wild-type protein. Our results show that Met side chains in the NTD core facilitate structural plasticity, which tightens dimerization through shape-optimization of a mobile binding interface.

## Results

**Unusual amino acid composition of MaSp NTDs.** In search for the mechanism of conformational change we analyzed the amino acid composition of aligned sequences of MaSp NTDs, published previously[8]. The amino acid composition shows striking pecu-liarities. Alanine and serine are most abundant with a combined content of ~30%. Only very few charged side chains are present

and the domain is unusually rich in Met. We focussed specifically on the NTD of MaSp1 from the nursery web spider *E. australis* because this representative has been intensely investigated in the past[8,11–14,17]. The sequence contains 12.4% alanine, which is more than the average 8.1% alanine found in proteins from all three domains of life (*Bacteria*, *Archaea* and *Eukaryota*)[18]. Ala-nine is known to stabilize helical secondary structure[19] and is presumably important for the structural integrity of the five-helix bundle. The NTD contains only 9% charged side chains, which is little compared to the average content of 25% charged side chains typically found in proteins[18]. Water-solubility of proteins is commonly provided by charged and hydrophilic side chains located on a surface. Water-solubility of the NTD appears to be facilitated by its unusually high content of 16.1% serine, which is a hydrophilic amino acid that can compensate the lack of charges. The average content of serine in bacterial, archaeal and eukaryotic proteins, by contrast, is only 6.2%[18].

Met is commonly an infrequent amino acid with an average content of only 2.5% in proteins from across all phyla of life[18]. The content of Met in dragline silk is even lower (0.2–0.4%)[20]. It therefore took us by surprise to find that NTDs from MaSps contain a substantial number of Met residues (content of 7.4% Met in the sequence of MaSp1 NTD from *E. australis*, Fig. 1a). Met is unevenly distributed along a spidroin sequence and accumulates in the NTD. We analyzed the location of Met side chains in the NTD using sequence alignment and homology of available structural data[8,14–16]. We found that the majority of Met residues are buried in the hydrophobic core and/or involved in tertiary interactions of helices. Only few Met residues were solvent-exposed (Fig. 1a). The high abundance of Met in the domain core indicated an unknown structural and/or functional role of this side chain.

**Solution structure of a Met-depleted spidroin NTD.** We investigated the structural role of Met in the NTD of MaSp1 from *E. australis*. Visual inspection of the structure (PDB IDs: 2LPJ and 3LR2)[8,14] showed that six of a total of ten Met side chains of the 137-residue domain are located in the core and/or are involved in tertiary interactions. The branched aliphatic side chain of Leu has a similar hydrophobicity and size as Met and can therefore serve as a suitable replacement[21,22]. To investigate the potential structural, dynamic and energetic effects of the six core Met we replaced them simultaneously by Leu, yielding a construct that we termed L6-NTD. Even though mutations in protein cores are generally not well tolerated and often act destabilizing[23], we found that L6-NTD expressed well and at high yield. Far-UV circular dichroism (CD) spectroscopy showed that the helical secondary structure of the wild-type NTD (WT-NTD) was retained in the mutant (Fig. 1b). Using NMR spectroscopy, we determined the atomic-resolution solution structure of L6-NTD at pH 7.0, i.e., under conditions where the domain is a monomer (Fig. 1c, Supplementary Fig. 1, Supplementary Table 1, PDB ID: 6QJY). The solution structure showed that L6-NTD is a well-folded, five-helix bundle with no significant deviations from the structure of the WT-NTD. The helices of L6-NTD and WT-NTD have the same respective lengths and form the same tertiary interactions. The all-atom alignment of residues 9–129 (omitting the flexible, unstructured C-terminal and N-terminal tails) of the lowest-energy conformer of L6-NTD with that of the WT-NTD[14] yielded a heavy-atom RMSD of 1.2 Å. The backbone heavy-atom RMSD was 1.2 Å. Helix orientations and positions of mutated side chains superimposed well. The conformation of the con-served Trp, which is wedged in the center of the helix bundle[14], was also preserved. We concluded that replacement of the six core Met by Leu had no impact on the structure of the NTD.

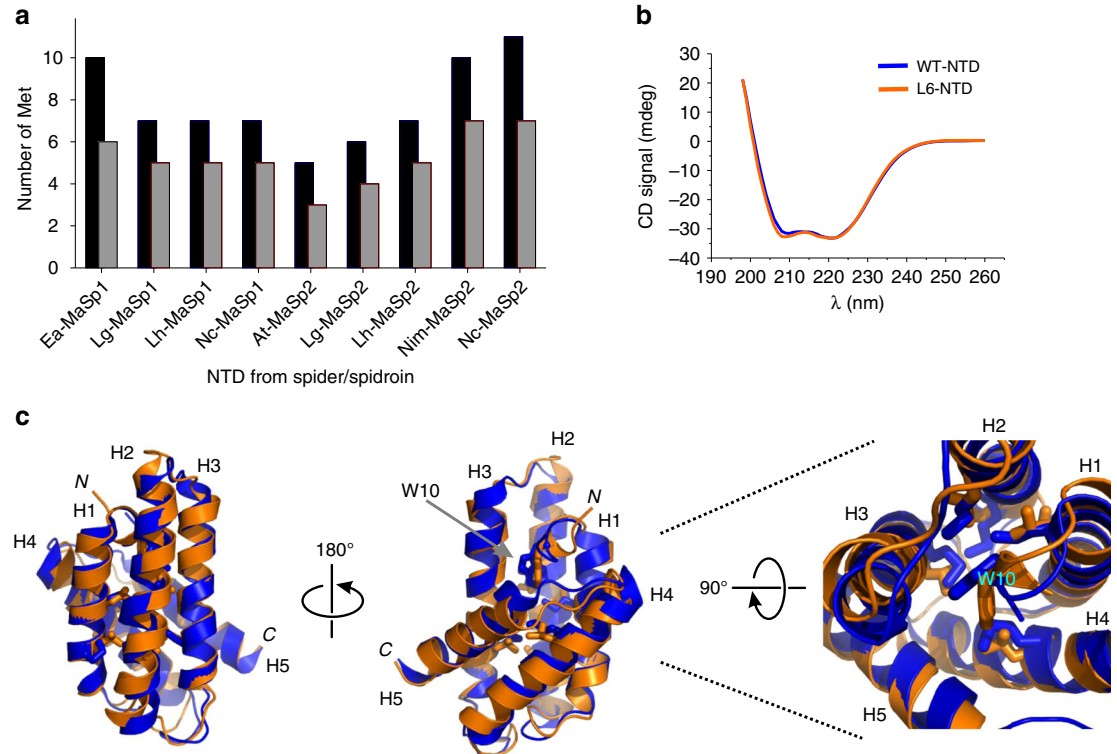

**Fig. 1** Met content in spidroin NTDs and solution structure of L6-NTD. **a** Number of Met residues in homologous, ~135-residue long NTD sequences of MaSp1 and MaSp2 from spider species Ea *Euprosthenops australis*, Lg *Latrodectus geometricus*, Lh *Latrodectus hesperus*, Nc *Nephila clavipes*, At *Argiope trifasciata*, and Nim *Nephila inaurata madagascariensis* (black bars). The number of Met residues in core position and/or involved in tertiary interaction are shown as gray bars. **b** Far-UV CD spectrum of WT-NTD (blue) and of L6-NTD (orange). **c** Solution NMR structure of L6-NTD (PDB ID: 6QJY) (orange) aligned with the structure of WT-NTD (PDB ID: 2LPJ) (blue). The side chains of the six Met residues that are in core position and/or involved in tertiary interaction in WT-NTD and are mutated to Leu in L6-NTD are highlighted as blue and orange sticks. The side chain of the conserved Trp (W10), helices 1-5 (H1-H5) and N-terminus (*N*) and C-terminus (*C*) are indicated. Source data of Fig. 1b are provided as Source Data file

**L6-NTD folds at the speed limit**. In order to investigate the influence of the Met-to-Leu mutations on the mechanism and energetics of folding, we performed chemical and thermal denaturation experiments of L6-NTD under pH 7.0 solution conditions where the NTD is a monomer. Equilibrium denaturation data revealed a dramatic increase of stability of L6-NTD compared to WT-NTD (Fig. 2a–c, Table 1). The free energy of folding increased from $5.6 \pm 0.1$ kcal/mol of WT-NTD to $8.0 \pm 0.3$ kcal/mol of L6-NTD (mean value ± s.d. of denaturation data measured by CD and fluorescence). The equilibrium m-value ($m_{D-N}$) of L6-NTD was slightly reduced compared to the WT value (Table 1). Changes in m-value are known to correlate with changes in accessible surface area between native and denatured states[24]. Since the structures of native WT-NTD and L6-NTD overlay (Fig. 1c), the slightly lower m-value measured for L6-NTD indicates that its denatured state is slightly more compact than that of WT-NTD. The melting temperature $T_m$ increased from originally $61.0 \pm 0.1$ °C of WT-NTD to $81.0 \pm 0.2$ °C of L6-NTD (±s.e. from regression analysis). To test whether stabilization resulted from a single mutation or was an additive effect[25] we generated six cumulative point mutants that build up L6-NTD (L1: M20L; L2:M20L/M24L; L3: M20L/M24L/M41L; L4: M20L/M24L/M41L/M48L; L5: M20L/M24L/M41L/M48L/M77L; L6: M20L/M24L/M41L/M48L/M77L/M101L) and analyzed their thermal stabilities. We found a cumulative increase of melting temperature, which showed that the observed effect of stabilization was additive (Fig. 2c). Mutation M48L (L4) had little effect on stability of L3 and thus behaved differently (Fig. 2c, inset). The structure shows that the side chain of residue M48 is squeezed

between helix 2 and helix 3, and is located close to the protein surface. M48 appears to form a less well consolidated van-der-Waals interaction network than the other Met side chains probed, which may explain the lack of effect upon mutating this residue. On the other hand, Met is reported[26] to interact specifically with Trp side chains, which may cause a stabilizing effect. Removing this stabilizing interaction through mutation may be balanced by the added stability increment from substitution by Leu.

To gain insight into the origin of the enhanced stability of L6-NTD, we measured its kinetics of folding using stopped-flow Trp fluorescence spectroscopy (Fig. 2d, e). Our previous kinetic experiments performed on a set of homologous spidroin NTDs shows that WT-NTDs fold rapidly on a time scale of ~100 μs through a barrier-limited two-state transition[11]. Here we found that L6-NTD folded 40 times faster than the WT-NTD (Table 1). The rate constant of folding of L6-NTD, extrapolated to zero denaturant, was $k_f = 519,000 \pm 213,000$ s$^{-1}$ (±s.e. from regression analysis). Synergy of theory and experiment predicts that the speed of folding of a generic *N*-residue single-domain protein is limited to $\tau_f = N/100$ μs, which is referred to as the speed limit of folding[27]. Following this theory, the speed limit of folding of the 137-residue NTD is ~1.4 μs. For L6-NTD we determined $\tau_f = 1/k_f = 1.9 \pm 0.8$ μs, which corresponds to the theoretical speed limit. Besides the observed increase of $k_f$ we found that L6-NTD was stabilized by a 15-fold decrease of unfolding rate constant compared to the WT-NTD (Table 1). From $k_f$ and $k_u$ of WT-NTD and L6-NTD we estimated a decrease of the free energy barrier between the denatured and the transition state by ~2.2 kcal/mol, and an increase of the free energy barrier between

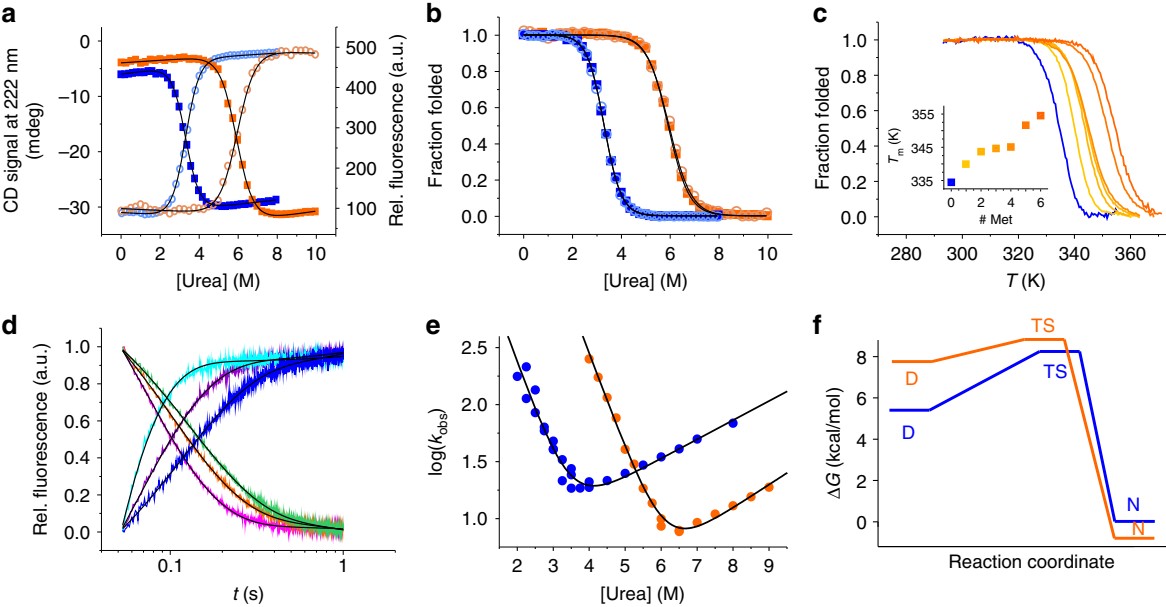

**Fig. 2** Equilibrium denaturation and folding kinetics of L6-NTD at pH 7.0. **a** Chemical denaturation of L6-NTD (orange) and WT-NTD (blue) measured using far-UV CD spectroscopy (light open circles) and Trp fluorescence at 325 nm (closed squares). Black lines are fits to the data using a thermodynamic model for a two-state equilibrium. **b** Data shown in **a** but normalized to the fraction of folded protein. Same symbol and color code applies. **c** Thermal denaturation of six cumulative Met-to-Leu mutants (blue to orange) measured using far-UV CD spectroscopy. Data are normalized to the fraction of folded protein. The increase of transition midpoint ($T_m$) rank-orders with the cumulative number of mutations (inset). **d** Kinetic transients of folding (cyan, violet, blue; 5.0, 5.5, 6.0 M urea, respectively) and of unfolding (green, orange, magenta; 6.5; 7.5, 8.5 M urea, respectively) of L6-NTD measured using stopped-flow Trp fluorescence spectroscopy. Signal amplitudes were normalized for reasons of clarity. Data were fitted using a mono-exponential decay function (black lines). **e** Observed rate constants (WT-NTD: blue, L6-NTD: orange) plotted versus concentration of urea (chevron plot). Data were fitted using a model for a barrier-limited two-state transition (black lines). **f** Free-energy profile of folding of WT-NTD (blue) and L6-NTD (orange) inferred from equilibrium and kinetic denaturation data. D Denatured state, TS transition state, and N native state are indicated. The free energy of the native state of WT-NTD was set as a reference state to zero kcal/mol. Source data are provided as Source Data file

---

**Table 1 Thermodynamic and kinetic data of folding of L6-NTD compared to WT-NTD**

|  | WT-NTD[a] | L6-NTD |
|---|---|---|
| $m_{D-N}^{CD}$ (kcal mol$^{-1}$ M$^{-1}$) | 1.69 ± 0.03 | 1.30 ± 0.05 |
| $m_{D-N}^{F}$ (kcal mol$^{-1}$ M$^{-1}$) | 1.69 ± 0.02 | 1.37 ± 0.03 |
| [urea]$_{50\%}^{CD}$ (M) | 3.30 ± 0.01 | 5.99 ± 0.02 |
| [urea]$_{50\%}^{F}$ (M) | 3.31 ± 0.01 | 5.95 ± 0.01 |
| $\Delta G_{D-N}^{CD}$ (kcal mol$^{-1}$) | 5.6 ± 0.1 | 7.8 ± 0.3 |
| $\Delta G_{D-N}^{F}$ (kcal mol$^{-1}$) | 5.6 ± 0.1 | 8.2 ± 0.2 |
| $k_f$ (s$^{-1}$) | 13,000 ± 6000 | 519,000 ± 213,000 |
| $k_u$ (s$^{-1}$) | 3 ± 1 | 0.2 ± 0.1 |
| $m_f$ (kcal mol$^{-1}$ M$^{-1}$) | 1.2 ± 0.1 | 1.13 ± 0.05 |
| $m_u$ (kcal mol$^{-1}$ M$^{-1}$) | 0.23 ± 0.03 | 0.30 ± 0.04 |
| $\beta_T$ | 0.84 ± 0.02 | 0.79 ± 0.09 |

$\beta_T$ denotes the Tanford $\beta$-value, calculated from $m_f$ and $m_u$ (ref. [17]). Errors of $\beta_T$ are propagated s.e. from regression analysis
[a]Equilibrium thermodynamic quantities are from ref. [17]; kinetic quantities are from ref. [11]. Errors are s.e. from regression analysis

---

the native and the transition state by ~1.6 kcal/mol, of L6-NTD compared to WT-NTD (Fig. 2f).

**Met-depletion of the NTD core impairs dimerization.** Conservation of structure and increase of stability suggested that the L6 mutant should retain functionality. Functionality of NTDs can be tested by measuring their ability to undergo pH-induced dimerization, in analogy to what happens in a spider's spinning duct[4,9,12,13,28]. NTDs at pH 6.0 are tight dimers that exhibit dissociation constants ($K_d$) in the low nanomolar range[11]. We characterized the equilibrium dimerization of L6-NTD and its response to changing solution conditions using size-exclusion chromatography in combination with multi-angle light scattering spectroscopy (SEC-MALS). At pH 7.0 and at 200 mM ionic strength, L6-NTD eluted homogenously as a monomer, as expected, similar to WT-NTD (expected molecular weight, $M_w$ = 14 kDa; measured $M_w$ = 14 kDa). At pH 6.0 and 60 mM ionic strength, however, the average molecular mass of L6-NTD measured by SEC-MALS was between that of a monomer and a dimer (expected $M_w$ = 28 kDa; measured $M_w$ = 20 kDa; Fig. 3a). The detected intermediate mass can be interpreted as that of a system in rapid, dynamic equilibrium between a monomer and a dimer. The intermediate $M_W$ indicated an elevated $K_d$ of L6-NTD compared to WT-NTD, which fell in the μM sample concentration range applied in this experiment. WT-NTD measured under the same conditions showed a molecular mass close to the value expected for a dimer (expected $M_w$: 28 kDa; measured $M_w$: 26 kDa, Fig. 3a). The measured value was only 7% below the expected value for a dimer. The discrepancy was little above the precision of the measurement (±3–5%) and may be explained by small amounts of salt (60 mM ionic strength) present in SEC experiments, which were required to reduce sticking of protein material to the column. Ions in solution shield electrostatics and induce dissociation of the WT-NTD[9,10,12]. High concentrations of salt lead to Debye–Hückel[29] screening of attractive electrostatic forces in the dimerization interface. This explains residual population of monomer in the dimer elution band and consequently a lower detected molecular mass. In order to strengthen

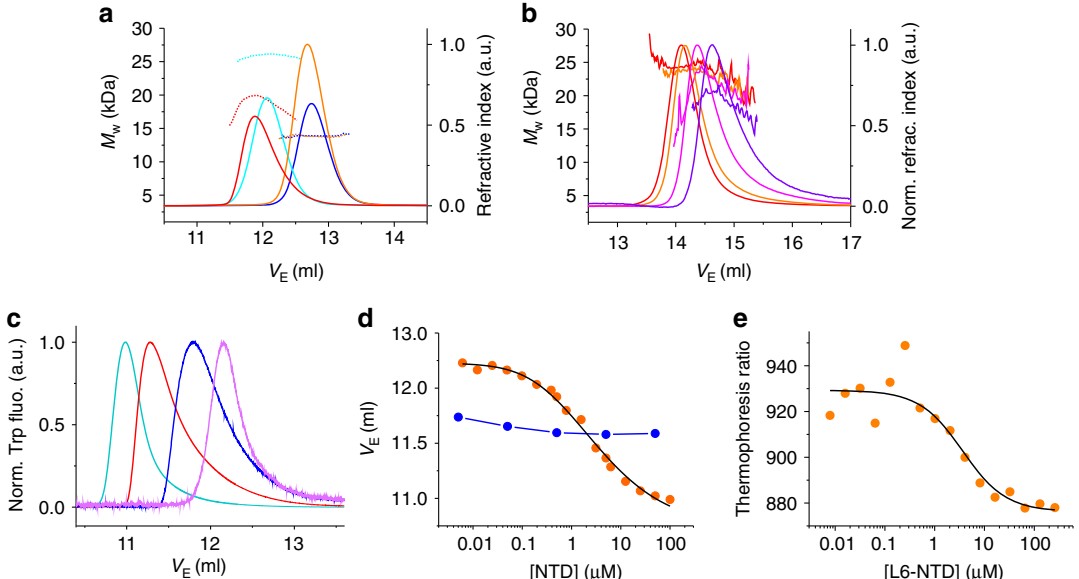

**Fig. 3** NTD dimerization assays. **a** SEC-MALS of L6-NTD and WT-NTD. SEC elution bands (solid lines) and corresponding average values of $M_w$ (broken lines) measured by MALS are shown. L6-NTD and WT-NTD measured at pH 7.0 (200 mM ionic strength) are shown in orange and blue. L6-NTD and WT-NTD measured at pH 6.0 (60 mM ionic strength) are shown in red and cyan. **b** SEC-MALS of 12 μM (red), 4 μM (orange), 1.2 μM (magenta), and 0.4 μM (blue) L6-NTD measured at pH 6.0 (8 mM ionic strength). Signals of elution bands are normalized to peak intensities. **c** High-resolution SEC of 100 μM (cyan), 6 μM (red), 0.8 μM (blue), and 0.05 μM (violet) L6-NTD measured using Trp fluorescence detection. Signals of elution bands are normalized to peak intensities. **d** $V_E$ of L6-NTD (orange) and WT-NTD (blue) measured at pH 6.0 (20 mM ionic strength) and at various protein concentrations using a high-resolution SEC. The black line is a fit to the data using a thermodynamic model for a monomer/dimer equilibrium. **e** Thermophoresis ratio of fluorescently modified L6-NTD-Q50C in presence of increasing concentration of L6-NTD (orange). The black line is a fit using a model for a binding isotherm. Differences in absolute peak values of $V_E$ of elution bands shown in panels (**a**–**c**) originate from the different SEC columns used in these experiments. Source data are provided as Source Data file

dimerization we reduced the ionic strength of the pH 6.0 buffer to 8 mM. Under these conditions, the measured molecular mass of L6-NTD was closer to that of a dimer (Fig. 3b). Dimerization of L6-NTD was thus apparently stabilized by electrostatic forces that are screened at high solution ionic strength, similar as previously observed for WT-NTD[9,10,12]. Upon progressive reduction of concentration of L6-NTD samples in SEC-MALS experiments we measured a decrease of molecular weight and an increase of elution volume ($V_E$), confirming that, despite low ionic strength, dimerization was still weak. The $K_d$ of L6-NTD was in the concentration range probed during measurement (i.e., $K_d$ in the μM range) (Fig. 3b). However, we were not able to obtain a full binding isotherm required to quantify $K_d$ because MALS detection was not sensitive enough to probe sub-μM protein concentrations. We therefore measured $V_E$ of dilute L6-NTD samples in pH 6.0 buffer using high-resolution SEC in combination with fluorescence detection, which had sub-μM sensitivity. An increase of L6-NTD concentration shifted the $V_E$ to lower values indicating formation of dimers (Fig. 3c). From concentration-dependent data we obtained a binding isotherm that fitted well to a thermodynamic model of dimerization, yielding a $K_d$ of 1.1 ± 0.2 μM (±s.e. from regression analysis) (Fig. 3d). This $K_d$ was three orders of magnitude higher than the value reported for WT-NTD ($K_d = 1.1 \pm 0.1$ nM)[11]. $V_E$ of WT-NTD at pH 6.0 did not change significantly over the probed protein concentration range because it was far above $K_d$[11] (Fig. 3d). However, there was a slight increase of $V_E$ of WT-NTD probed in the 10-nM concentration range, which may indicate the onset of dimer dissociation. The $V_E$ of the WT-NTD dimer was significantly higher than the value of the L6-NTD dimer, which indicated that the L6-NTD dimer had larger dimensions. Expansion of the L6-NTD dimer can be explained by loose association of subunits. To test if high-resolution SEC data reported reliably on $K_d$ of the dimer, we

conducted thermophoresis experiments as an alternative probe for dimerization (Fig. 3e). Using thermophoresis we obtained a $K_d$ of 3.6 ± 1.8 μM, which was in reasonable agreement with the value obtained by high-resolution SEC.

Next, we investigated the influence of individual Met side chains on size and stability of the NTD dimer by measuring dimerization of six cumulative Met-to-Leu point mutants using high-resolution SEC (mutations are detailed above in denaturation experiments). The obtained isotherms are shown in Fig. 4a–e. The $K_d$ increased progressively with increasing number of mutations. At the same time, the $V_E$ of the dimer decreased with increasing number of mutations (Fig. 4f). This showed that the dimensions of the dimer increased progressively with increasing number of mutations, which indicated progressive loosening of the complex.

**Met facilitates conformational changes upon dimerization.** Dimerization of the NTD is known to be associated with conformational changes. Helices that form the dimerization interface tilt and the conserved Trp (W10) moves out of its binding pocket to solvent-exposed position[14]. This is evident from the quenching and red-shift of Trp fluorescence emission, effects that are conserved across NTDs from various glands and species[8,9,11,13,14]. We found that the L6-NTD lacked these fluorescence characteristics. There was no red-shift and only minor quenching of Trp fluorescence emission upon dimerization (Fig. 5a). The observation indicated that in L6-NTD Trp W10 remained in buried position both in the monomeric and in the dimeric state. Fluorescence characteristics changed gradually with increasing number of Met-to-Leu mutations (Fig. 5b), similar as observed for the change of stability and of $K_d$ (Figs. 2c and 4f). Full conformational change of Trp thus required the presence of several core Met.

We measured and analyzed $^1$H, $^{15}$N HSQC NMR spectra of L6-NTD and WT-NTD recorded in both the monomeric and the

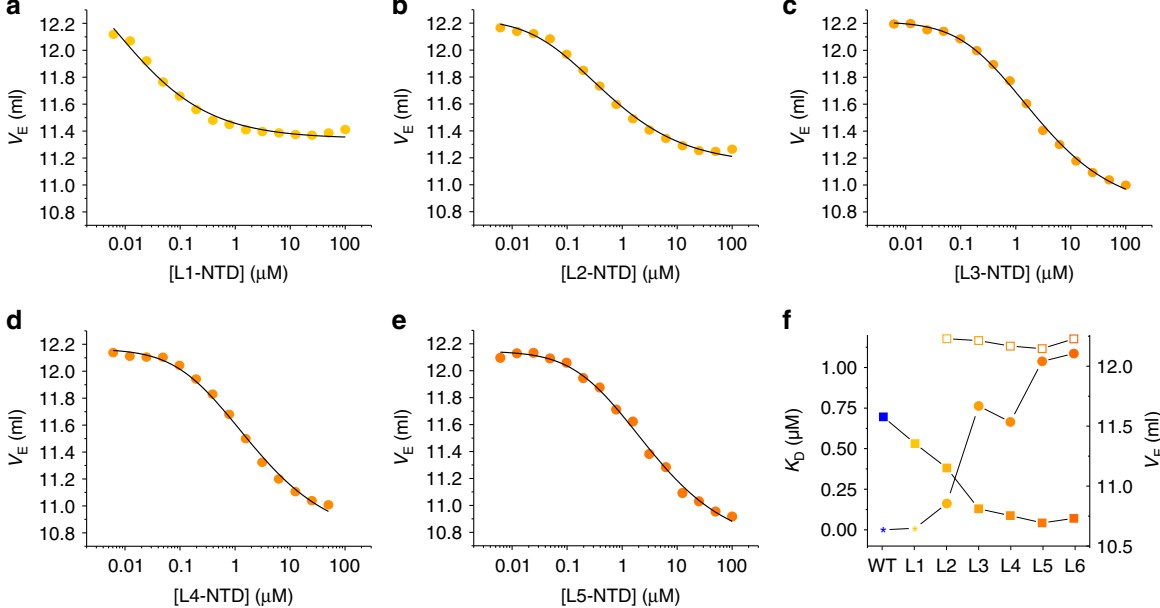

**Fig. 4** Dimerization of cumulative Met-to-Leu mutants. **a–e** $V_E$ plotted versus protein concentration of cumulative Met-to-Leu mutants L1 to L5 measured using high-resolution SEC. The black lines are fits to the data using a thermodynamic model for a monomer/dimer equilibrium. **f** Values of $K_d$ (spheres) and end-point values of $V_E$ (squares) of cumulative Met-to-Leu mutants (L1 to L6) and of WT-NTD measured using high-resolution SEC. The $K_d$ of WT-NTD is the value determined from kinetics reported in ref. [11]. The $K_D$ of L1 is an estimate because of the non-defined monomer baseline. Closed and open squares are end-point $V_E$ values of dimer and monomer, respectively. The end-point values of $V_E$ of the WT-NTD monomer and of the L1-NTD monomer could not be determined because their $K_d$ was too low to enter baseline region. Source data are provided as Source Data file

dimeric states. The full $^1H$, $^{15}N$ HSQC NMR data set is shown in Supplementary Fig. 2. In agreement with our results from fluorescence spectroscopy, $^1H$, $^{15}N$-Trp chemical shifts of L6-NTD measured by NMR spectroscopy showed only minute perturbations upon change of solution pH from 7.0 to 6.0, which was in stark contrast to what we observed for the WT-NTD (Fig. 5c). The NMR data showed that the environment of W10 in L6-NTD remained essentially unchanged upon dimerization, contrasting the substantial changes of the environment of W10 in WT-NTD. Figure 5d shows the respective differences in chemical shifts of all assigned residues in WT-NTD and L6-NTD between pH 7.0 and pH 6.0 (full HSQC spectra are provided as Supplementary Fig. 2). Residues of helices H1, H4, and H5 of L6-NTD could be reliably assigned at pH 6.0. However, assignment of helices H2 and H3 of L6-NTD was complicated by line broadening caused by dynamic exchange on the intermediate time scale, in agreement with the reduced dimer affinity. The observation supports our finding that mutation of Met to Leu in the domain core has profound consequences both for dynamics and function of the protein and prevents the dimer interface to adopt a conformation suitable for high-affinity dimerization. A loose L6-NTD dimer was found in the high-resolution SEC experiments described above (Fig. 4f). The loose dimer presumably shows rapid inter-molecular interaction dynamics between subunits that may enhance NMR line broadening. However, analysis of the many assigned residues showed that chemical shift changes upon dimerization covered the entire sequence and were in general substantially larger for WT-NTD compared to L6-NTD (Fig. 5d). The result was in agreement with strong dimer interactions and more extensive conformational changes in WT-NTD, which are apparently blocked in L6-NTD. In L6-NTD, the majority of chemical shift changes of residues that are part of the dimerization interface resulted from weak dimer interactions. Interestingly, we found pronounced chemical shift changes of resonances of glycine residues located in loops connecting helices of WT-NTD upon

dimerization. These chemical shift changes were absent in L6-NTD (Fig. 5e). Since the glycine-rich NTD loops are not involved in dimerization, their significant chemical shift changes upon dimerization in WT but not in L6-NTD indicated that Met promotes conformational changes that are remote from the core and the dimerization interface.

**Met drives native-state dynamics of the NTD monomer.** Having established that Met enables structural changes required for dimerization, we set out to investigate the influence of core Met on native-state conformational dynamics of the NTD monomer. For insights into per-residue dynamics and changes in solvent-accessibility of residues in the L6-NTD structure in comparison to WT-NTD, we performed NMR-based hydrogen/deuterium exchange experiments (Fig. 6a). We found that proton-to-deuterium exchange in L6-NTD was on average ~10-fold slower compared to WT-NTD, i.e., L6-NTD was overall less dynamic and the exchange-competent conformations were less frequently accessible than in WT-NTD. For L6-NTD, H/D exchange was significantly slower not just at the positions of mutation but throughout all five helices, i.e., through residues 18–28, 41–51, 72–81, 98–107, and 118–128 (Fig. 6a, b). In contrast, the local backbone dynamics on the fast nanosecond-to-picosecond time scale were not influenced by the mutations, as indicated by heteronuclear NOE measurements (Fig. 6c, d). Protein dynamics on the nanosecond-to-picosecond time scale generally report on local events, while dynamics on time scales of microseconds (μs) and slower reflect collective conformational motions[30]. Met residues in the core of the NTD thus appeared to facilitate collective motions rather than local, uncoupled flexibility.

To probe conformational motions on the μs time scale we applied photoinduced electron transfer (PET) in combination with fluorescence correlation spectroscopy (PET-FCS), a technique that measures conformational dynamics from single-molecule

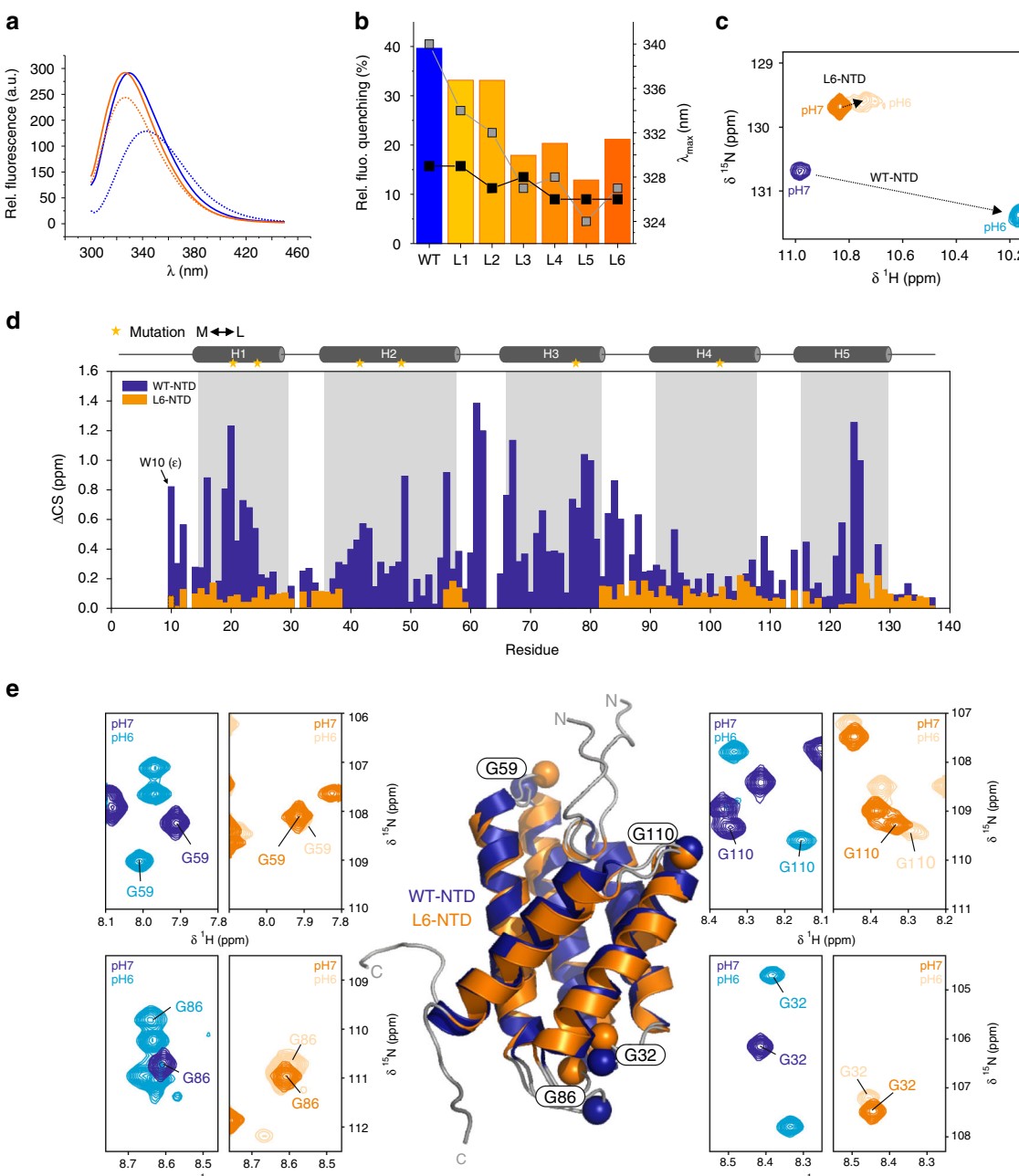

**Fig. 5** Native-state conformational changes of the NTD. **a** Trp fluorescence spectra of WT-NTD (blue) and L6-NTD (orange) measured at pH 7.0 and 200 mM ionic strength (solid lines), and at pH 6.0 and 8 mM ionic strength (dashed lines). **b** Quenching of Trp fluorescence emission of cumulative Met-to-Leu mutants upon change of solution condition from pH 7.0 (200 mM ionic strength) to pH 6.0 (8 mM ionic strength) (colored bars). The wavelengths of maximal fluorescence emission at pH 7.0 (gray squares) and at pH 6.0 (black squares) is shown for each mutant. **c** $^1$H, $^{15}$N-HSQC NMR spectra detailing the side chain amide signal of W10 of L6-NTD and WT-NTD in pH 7.0 buffered solution and 200 mM ionic strength (orange and blue, respectively) and in pH 6.0 and 8 mM ionic strength (light orange and cyan, respectively). **d** NMR chemical shift (CS) differences of assigned residues in the WT-NTD (blue) and L6-NTD (orange) upon change of solution pH from 7.0 to 6.0. **e** $^1$H, $^{15}$N-HSQC NMR signals of NTD loop residues. Close up of the HSQC spectral regions of glycine residues G59, G86, G110, and G32 located in loop segments connecting helices of WT-NTD and L6-NTD recorded at pH 7.0 and at pH 6.0 are shown. The location of residues on the aligned structures of WT (blue) and L6-NTD (orange) is highlighted as spheres. Source data are provided as Source Data file. Source data shown in panel **e** are provided in the BMRB (entry: 27683)

fluorescence fluctuations[17,31,32]. We probed dynamics of the conserved Trp residue (W10), which moves in and out of the five-helix bundle[17]. To this end, we modified the side-chain thiol of an engineered cysteine introduced at the N-terminus, vicinal to W10 (mutant G3C), using the thiol-reactive fluorophore AttoOxa11 (Fig. 6e). Motion of W10 from buried to solvent-exposed position lead to contact-induced quenching of the

fluorescence label via PET. PET-FCS autocorrelation functions recorded from AttoOxa11-modified WT-NTD and L6-NTD showed three decays. One decay was on the millisecond (ms) time scale and caused by molecular diffusion of the NTD through the detection focus. Two additional decays on the sub-ms time scale were caused by PET fluorescence fluctuations that reported on Trp conformational change (Fig. 6f). The sub-ms decays were

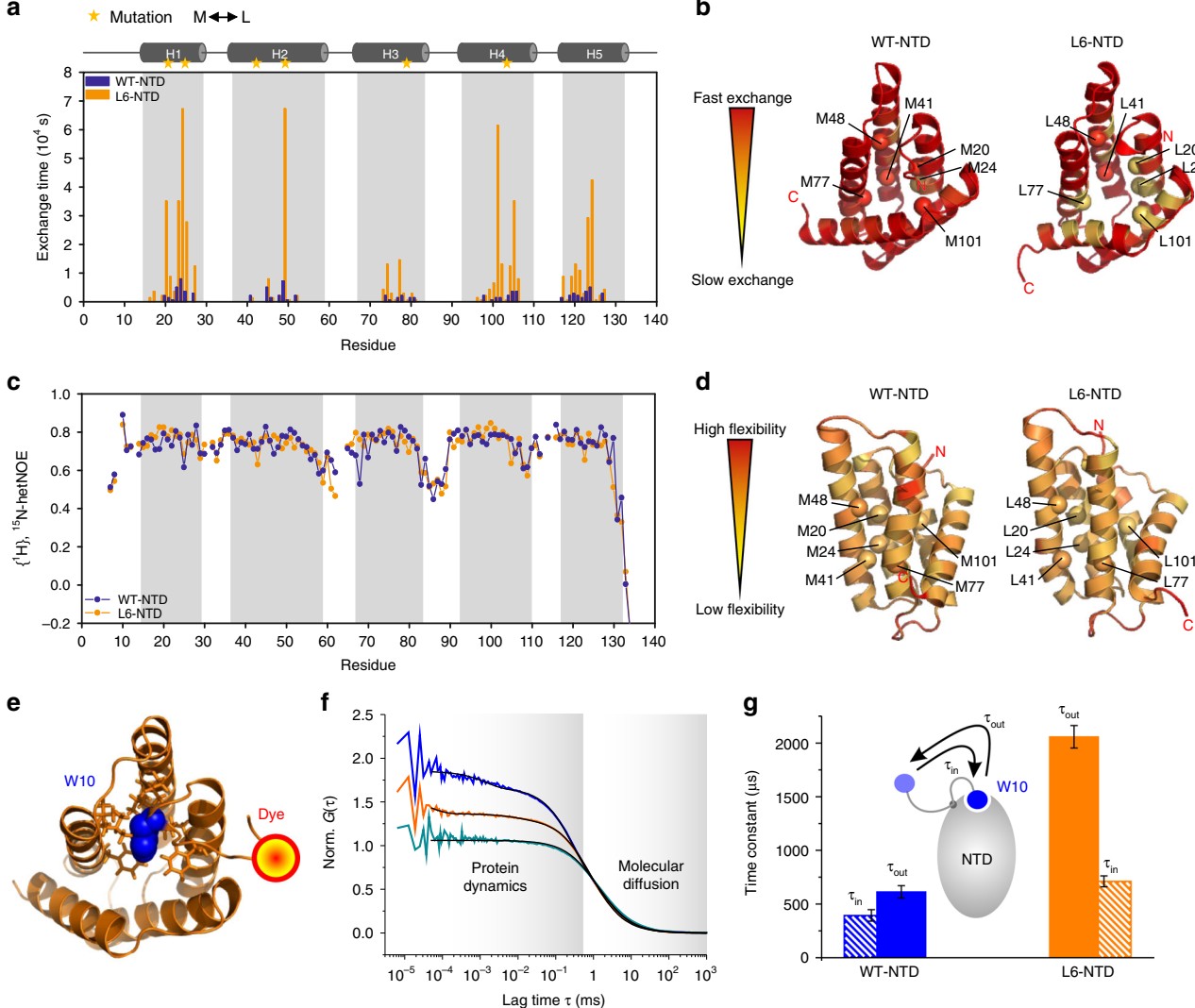

**Fig. 6** Native-state dynamics of L6-NTD probed by NMR spectroscopy and PET-FCS. **a** NMR H/D exchange experiments. Amide H/D exchange time of L6-NTD (orange) and WT-NTD (blue) plotted versus residue number. Positions of helices and mutations are indicated on top of the panel. **b** Exchange times imposed on the structure of WT-NTD (PDB ID: 2LPJ) and of L6-NTD (PDB ID: 6QJY). **c** {$^1$H},$^{15}$N-hetNOE values of WT-NTD (blue) and L6-NTD (orange) backbone amides plotted versus residue number. Lower values indicate higher flexibility. **d** {$^1$H},$^{15}$N-hetNOE values imposed on the structure of WT-NTD (PDB ID: 2LPJ) and L6-NTD (PDB ID: 6QJY), respectively. **e** Reporter design for PET-FCS. Top view of the structure of L6-NTD (PDB ID: 6QJY) indicating the position of the AttoOxa11 fluorescence label (red sphere) attached to the N-terminus (G3C). The side chain of the conserved Trp W10 (blue sphere), which quenches fluorescence of AttoOxa11 upon solvent-exposure, is highlighted. Trp is buried in a network of tertiary interactions (side chains shown as sticks). **f** PET-FCS autocorrelation functions ($G(\tau)$) recorded from WT-NTD and L6-NTD (blue and orange) modified with AttoOxa11 at position G3C. Data of L6-NTD modified with AttoOxa11 at position Q50C served as a control (cyan). Black lines are fits to the data using a model for translational diffusion of a globule containing two single-exponential relaxations. The control sample was described by a model for translational diffusion only. **g** Time constants of Trp moving out of the core ($\tau_{out}$, solid bars) and into the core ($\tau_{in}$, patterned bars) of WT-NTD (blue) and L6-NTD (orange), calculated from PET-FCS data. Error bars are propagated s.e. from regression analysis. The comic at the center of the panel illustrates the conformational change and assigns $\tau_{out}$ and $\tau_{in}$. The NTD is illustrated as gray ellipsoid. Trp W10 (blue sphere) moves between buried and solvent-exposed positions. Source data are provided as Source Data file

dominated by a single-exponential kinetic phase with a relaxation time constant $\tau_1 = 240 \pm 17$ μs and an amplitude $a_1 = 0.64 \pm 0.03$ measured for WT-NTD, and $\tau_1 = 529 \pm 19$ μs and $a_1 = 0.35 \pm 0.01$ measured for L6-NTD (±s.e. from regression analysis). The second kinetic phase had $\tau_2 = 2 \pm 1$ μs and $a_2 = 0.19 \pm 0.04$ measured for WT-NTD; and $\tau_2 = 4 \pm 1$ μs and $a_2 = 0.05 \pm 0.01$ measured for L6-NTD. The observed reduction of amplitude and increase of relaxation time constant of the main kinetic phase indicated that the mobility of W10 was reduced in L6-NTD compared to WT-NTD. From $\tau_1$ and $a_1$ we estimated[33] the microscopic time constants of Trp conformational change (Fig. 6g). The time

constant of W10 moving out of the binding pocket, as happens upon dimerization, was substantially larger for L6-NTD compared to WT-NTD ($\tau_{out}^{WT} = 0.61 \pm 0.06$ ms, $\tau_{out}^{L6} = 2.1 \pm 0.1$ ms; Fig. 6g). Thus, core Met in WT-NTD accelerated the release of Trp W10 from buried to solvent-exposed position.

## Discussion

Synthesis of silk fibers by web spiders involves dimerization of the spidroin NTD, which polymerizes protein building blocks[4,9,10]. The seemingly simple process underlies a complex multi-step

mechanism[4,12,13] triggered by a gradual change of pH along the spinning duct[28]. A gradual mechanism makes physiological sense because soluble spidroins need to orient and align during their passage through the spinning duct before inter-molecular interactions consolidate. Sequential protonation gives rise to early, attractive long-range electrostatic forces between pairs of NTDs. In the early complex, tight dimerization is prevented by a mismatch of shape of the dimerization interface[4,14], i.e., the subunits have the wrong conformation. Late rearrangement of helices adapts the interface and tightens binding. What is the mechanism behind these conformational changes and what is their energetic contribution to dimerization? Our study provides answers to these questions.

The shape of a protein surface is determined by the shape of the protein hydrophobic core[23]. Side chains in core position are commonly tightly packed and essentially immobile. Their interaction network resembles a jigsaw puzzle[34] where side chains are in extensive van-der-Waals contact and in low-energy conformation[23]. Structures of NTDs show a common core that is densely packed with hydrophobic side chains[8,14–16]. But we found that the rare[18,20] amino acid Met was unusually over-represented (Fig. 1a). This accumulation of Met may not be seen as a surprise because the amino acid has similar hydrophobicity and size compared to other aliphatic side chains frequently present in the core region of proteins[21,22]. However, we made surprising observations concerning the role of Met in structure and stability of the domain and revealed an intriguing functional role of its side chain, as we discuss below.

In protein engineering experiments the replacement of core side chains is usually avoided because it often leads to considerable destabilization of the fold[23,35]. Yet, we found that the simultaneous replacement of six core Met in the NTD yielded a well-folded protein that was substantially more stable than the wild type (~50% increase of $\Delta G$; Table 1). Extensive modification of the domain core and the resulting change of stability would typically suggest a change of structure. Surprisingly, the fold of L6-NTD was fully preserved with no noticeable deviations from the wild type (Fig. 3c). Stabilization and conservation of structure can be expected to preserve function. But dimerization of L6-NTD was dramatically impaired: its $K_d$ was ~1000-fold higher than the value of the wild type. The NTD thus emerges as yet another example of a protein where residues at positions of functional importance impair stability[36]. But what exactly does Met do to the NTD to enhance dimerization at the expense of stability?

We found that side chains of core Met facilitate motion of secondary and tertiary structure. This was evident from NMR hydrogen-exchange and PET-FCS experiments, which revealed that dynamics of the Met-depleted domain were substantially slower than dynamics of the Met-rich domain (Fig. 6). Models of protein-protein association suggest that native-state dynamics transiently populate activated states, which are competent to bind[37]. The NTD likely gains access to such activated states through Met-driven conformational changes, which prime the domain for tight dimerization. But how does Met induce such dynamics?

Met is unique among all 20 natural amino acids in that its side chain is highly flexible[38]. This unusual flexibility originates from a low energy barrier to rotation around the thioether bond[38]. Met positioned on a protein surface can make the surface ductile and thereby facilitate promiscuous binding[38–40]. Recently, reversible oxidation of solvent-exposed Met side chains has been reported to promote protein phase transition[41]. Oxidation of the thioether sulfur changes its polarity, but presumably also stiffens the side chain. Our results show that Met side chains in the core of the NTD transfer their flexibility on large parts of the structure and

thereby malleablize it (Fig. 6). The lack of Met in L6-NTD blocked conformational changes required for tight dimerization. However, the electrostatics of the domain remained unchanged. L6-NTD can thus serve as a model for the elusive intermediate state formed along the pathway of multi-step dimerization[13], in which early changes of surface electrostatics through protonation precede late conformational change. We can dissect the energetic contributions from electrostatics and conformational change. The total free energy of dimerization, which contains both contributions, can be calculated from the equilibrium dissociation constant of WT-NTD ($K_d = 1.1 \pm 0.1$ nM)[11]: $\Delta G_{WT} = -RT\ln(K_d) = -12.3 \pm 0.1$ kcal/mol (±propagated s.e. from regression analysis). The electrostatic contribution can be calculated from the $K_d$ of L6-NTD ($K_d = 1.1 \pm 0.2$ µM) where conformational changes are blocked and the domains are loosely held together by electrostatic forces: $\Delta G_{elec} = -RT\ln(K_d) = -8.2 \pm 0.1$ kcal/mol. The portion of the free energy resulting from conformational change is thus: $\Delta G_{conf} = \Delta G_{WT} - \Delta G_{elec} = -4.1 \pm 0.1$ kcal/mol, which represents a substantial fraction of the total free energy of dimerization.

We found that Met-to-Leu mutations accelerated folding and strongly stabilized the NTD monomer. Folding of L6-NTD was ultrafast and at the theoretical speed limit[27] (Table 1). Its folding time constant was close to one µs, which places L6-NTD amongst the fastest folding protein domains of this size[27]. Acceleration of folding is explained by changes of structure and/or interactions in the denatured or transition state of folding. Those effects are generally hard to analyze because both states are elusive to experimental observation[42]. It has been suggested that the large polarizability of the sulfur atom of Met renders the side chain more sticky at van der Waals contact compared to other hydrophobic side chains[38]. Met may thus induce stronger interactions in the denatured state and thereby reduce its free energy. Mutation of Met abolishes such interactions and, in turn, increases the free energy of the denatured state, which brings it closer to the transition state. A reduced free energy gap between denatured and transition state will increase the rate constant of folding (Fig. 2f). On the other hand, replacement of Met by Leu slowed unfolding (Table 1), which shows that these mutations stabilized the native state. Leu has a branched aliphatic side chain that perhaps forms more interlocked and extensive van der Waals interactions in the native state compared to the linear Met side chain, which can explain the observed stabilization.

Stable proteins are more capable to evolve new or improved functions compared to unstable ones because they can better tolerate mutations[43]. Reconstructed ancestors are consequently more stable than their modern descendants[44]. We hypothesize that the stable, Leu-rich core of L6-NTD resembles an early ancestor of the highly evolved Met-rich core. Phylogenetic studies show that Met is among the most frequently gained amino acids in protein evolution[18]. DNA codons of Met, Leu and isoleucine (Ile) differ by only one nucleobase. Evolutionary transformation of Leu or Ile to Met requires thus only single nucleotide exchange and has higher probability than transformation to other amino acids. Alignment of homologous sequences[16] shows that Met is overrepresented in NTDs of major and minor ampullate, as well as of aciniform spidroins. Interestingly, Met, Leu and Ile often share the same sequence positions (Fig. 7). In fact, the sum of Met, Leu and Ile is conserved in each NTD and the lack of Met in NTDs from tubuliform or cylindrical silk is compensated by Leu or Ile (Fig. 8). This supports our hypothesis that a Leu/Ile-rich NTD is an early ancestor of the Met-rich NTD. It is tempting to speculate about the influence of Met in the spidroin NTD on the mechanical properties of the different silk types they constitute. Silks containing Met-rich, high-affinity NTDs should exhibit higher strength compared to silks containing Met-depleted, low-affinity NTDs. Major and minor ampullate, as well as aciniform

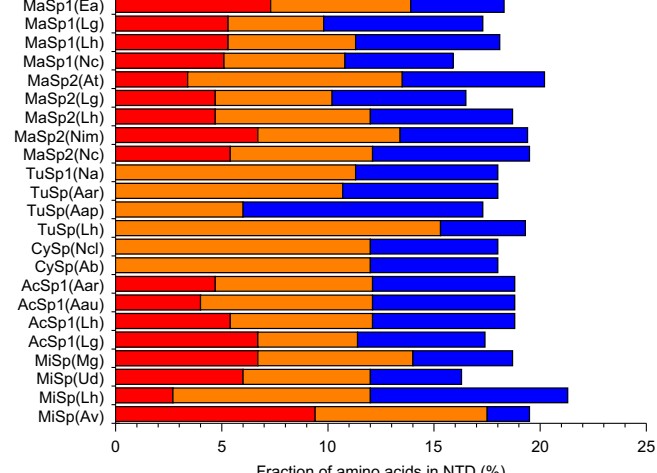

**Fig. 7** Sequence alignment of homologous spidroin NTDs. Identical (*), very similar (:) and similar (.) side chains are indicated at the bottom. Met (M), Leu (L), and Ile (I) residues are highlighted in red, orange and blue, respectively. Abbreviations of spider species and spidroins: Ea *Euprosthenops australis*, Lg *Latrodectus geometricus*, Lh *Latrodectus hesperus*, Nc *Nephila clavipes*, At *Argiope trifasciata*, and Nim *Nephila inaurata madagascariensis*, Na *Nephila antipodiana*, Aar *Argiope argentata*, Aap *Agelenopsis aperta*, Ncl *Nephila clavata*, Ab *Argiope bruennichi*, Aau *Argiope aurantia*, Mg *Metepeira grandiosa*, Ud *Uloborus diversus*, Av *Araneus ventricosus*; MaSp major ampullate spidroin, CySp and TuSp cylindrical silk/ tubuliform spidroin, AcSp aciniform spidroin, and MiSp minor ampullate spidroin

silks, which are used by web spiders to build a web and for wrapping prey, are reported to exhibit high toughness[45]. Indeed, their spidroin NTDs show high content of Met (Fig. 8). Tubuliform silk, on the other hand, is a flocculent silk used to build the egg case, same as cylindrical silk. It is less tough[45] and its spidroin NTDs contain no Met (Fig. 8).

We learned from web spiders that the accumulation of Met in a protein hydrophobic core induces structural plasticity and enables dynamical excursions to conformational states, which, in the case of the spidroin NTD, facilitate tight binding. Mobilization of a hydrophobic core can modulate a protein surface where function is defined and optimize it. We may thus be able to improve or change the functionality of proteins by engineering hydrophobic cores to become more dynamic.

## Methods

**Protein expression, mutagenesis and modification**. The synthetic gene (GeneArt, Thermo Fisher Scientific) of the MaSp1 NTD from *E. australis*, was cloned into a pRSETA vector (Invitrogen, Thermo Fisher Scientific) containing an N-terminal His$_6$-tag followed by a thrombin cleavage site for proteolytic removal of

**Fig. 8** Frequencies of Met (red), Leu (orange), and Ile (blue) in homologous spidroin NTD sequences (specified in Fig. 7)

the tag[11]. For site-directed mutagenesis experiments the QuikChange mutagenesis protocol (Stratagene) was used. We generated L6-NTD through cumulative single-point Met-to-Leu mutants. Met sequence positions mutated to Leu were 20, 24, 41, 48, 77, and 101. NTDs and mutants thereof were overexpressed in *Escherichia coli* C41 (DE3) bacterial cells and isolated from clarified cell lysate using affinity chromatography on a Ni-Sepharose 6 Fast-Flow column (GE Healthcare), followed by proteolytic cleavage of the His$_6$-tag using thrombin from bovine plasma (Sigma-Aldrich)[11]. Proteins were purified using size exclusion chromatography (SEC) on a Superdex 75 column (GE Healthcare) in 200 mM ammonium bicarbonate, pH 8.0. Purity of isolated protein was confirmed using SDS-PAGE. Pooled protein fractions from SEC were lyophilized. For thermophoresis and PET-FCS experiments, the single-point cysteine mutants Q50C and G3C were modified with the thiol-reactive maleimide derivative of the oxazine fluorophore AttoOxa11 (Atto-Tec). A 5-fold molar excess of fluorophore was added to the Cys mutant dissolved in 50 mM 3-(*N*-morpholino)propanesulfonic acid (MOPS), pH 7.5, containing 6 M guanidinium chloride and a 10-fold molar excess of tris(2-carboxyethyl)phosphine (TCEP) to prevent thiol oxidation. The labeling reaction was carried out for 2.5 h at 298 K. Labeled protein was isolated using SEC on a Sephadex G-25 column (GE Healthcare).

For NMR studies, cells were grown in minimal media supplemented with $^{15}$N ammonium chloride and $^{13}$C glucose as the sole nitrogen and carbon sources[46]. Isotope labeled protein was purified as described above.

**Far-UV CD spectroscopy.** Far-UV CD spectroscopy was carried out using a Jasco J-815 CD spectrometer equipped with a Peltier thermocouple. Sample temperature was set to 298 K throughout all experiments, except for thermal denaturation experiments where a temperature ramp of 1 K/min was applied. 10 μM protein samples were measured in a 1 mm path-length cuvette (Hellma). Chemical and thermal denaturation was monitored at 222 nm, i.e., the wavelength of maximal amplitude of α-helix secondary structure. Chemical denaturation was performed by manual titration between 0 and 8 M Urea in 50 mM MOPS, pH 7.0, with the ionic strength adjusted to 200 mM using potassium chloride. Thermal denaturation was carried out in 50 mM phosphate buffer, pH 7.0, with the ionic strength adjusted to 200 mM using potassium chloride.

**Steady-state fluorescence spectroscopy.** Trp fluorescence spectroscopy was carried out using a Jasco FP-6500 spectrometer. In chemical denaturation experiments Trp fluorescence emission intensities were recorded from 10 μM protein samples measured in a 10 mm path-length cuvette (Hellma). Spectra were recorded from 100 μM protein samples in either 50 mM phosphate, pH 7.0, with the ionic strength adjusted to 200 mM using potassium chloride (monomer conditions) or in 20 mM MES, pH 6.0 and 8 mM ionic strength (dimer conditions). The high protein concentration was applied to ensure dimerization of Met-depleted NTD mutants. Sample temperature was controlled using a Peltier thermocouple set to 298 K throughout all experiments. Chemical denaturation experiments were conducted under the same solution conditions as described above for CD spectroscopy.

**Stopped-flow fluorescence spectroscopy.** Kinetics of folding and unfolding were measured at 298 K using a SFM-2000 stopped-flow fluorescence spectrometer (BioLogic) equipped with a 280-nm diode as excitation source, monitoring Trp fluorescence emission of the NTD. The temperature was adjusted using a circulating water bath. One micromolar protein samples were prepared in 50 mM MOPS, pH 7.0, with the ionic strength adjusted to 200 mM using potassium chloride, containing either zero or six molar urea. All samples were filtered through 0.2 μm syringe filters before measurement and rapidly mixed into urea solutions of varying concentration applying a volumetric mixing ratio of 1:10 using the stopped-flow machine.

**SEC-MALS and high-resolution SEC experiments.** SEC-MALS was carried out using a Superdex-75 HR10/300 or a Superdex-200 HR10/300 analytical gel filtration column (GE Healthcare) run at 0.5 ml min$^{-1}$ in either 50 mM phosphate, pH 7.0, with the ionic strength adjusted to 200 mM using KCl (monomer conditions) or in 20 mM MES, pH 6.0, with the ionic strength adjusted to 60 mM using KCl or in 20 mM MES, pH 6.0 at 8 mM ionic strength (dimer conditions). The sample loading concentration reduced by 10-fold in the detection flow-path because of sample dilution on the column. Stated concentrations represent the maximal level during the experiment. At the edge of the peaks concentrations will be lower than this. The elution was followed in a standard SEC-MALS format using SEC UV absorbance, light scattering intensity measured with a Wyatt Heleos II 18 angle instrument, and finally excess refractive index using a Wyatt Optilab rEX instrument. The Heleos detector 12 was replaced with a Wyatt's QELS detector for dynamic light scattering measurements. Protein concentration was determined from the excess differential refractive index based on 0.186 RI increment for 1 g/ml protein solution. Concentrations and observed scattered intensities at each point in the chromatograms were used to calculate the absolute molecular mass from the intercept of the Debye plot, using Zimm's formalism as implemented in Wyatt's ASTRA software.

High-resolution SEC experiments were conducted using a Jasco HPLC work station with fluorescence detection equipped with a Superdex-75 Increase 10/300 column (GE Healthcare). SEC was run at a flow rate of 0.8 ml/min and in 20 mM MES, pH 6.0, with the ionic strength adjusted to 20 mM using potassium chloride. $V_E$ of NTD constructs, which were applied at various concentrations, was determined from the peaks of elution bands detected measuring the native Trp fluorescence signal. Native Trp fluorescence was excited at 280 nm and emission intensities were recorded at 330 nm.

**Thermophoresis experiments.** Microscale thermophoresis was measured using a Nanotemper Monolith instrument equipped for fluorescence excitation in the far-red spectral range. Forty nanomolar samples of labeled AttoOxa11-modified L6-NTD mutant Q50C were incubated with increasing concentrations of unlabeled L6-NTD and loaded into standard, untreated capillaries (Nanotemper). After localization of capillaries, thermophoresis was performed at 50% heating power and at 50% illumination intensity. The data were expressed as the ratio of fluorescence intensities before and at the end of 30 s of heating when the new, stable equilibrium intensities had been achieved.

**PET-FCS.** PET-FCS was carried out on a confocal fluorescence microscope setup[17] (Zeiss Axiovert 100 TV) equipped with a diode laser emitting at 637 nm (Coherent Cube) and a high numerical aperture oil-immersion objective lens (Zeiss Plan Apochromat, 63×, NA 1.4). The average laser power was 400 μW before entering the back aperture of the microscope lens. The fluorescence signal was shared by two fiber-coupled avalanche photodiode detectors (APDs; Perkin Elmer, SPCM-AQRH-15-FC). Signals of APDs were recorded in the fast cross-correlation mode using a digital hard-ware correlator device (ALV 5000/60 × 0 multiple tau digital real correlator). One nanomolar sample of fluorescently modified NTDs were prepared in 50 mM phosphate buffer, pH 7.0, with the ionic strength adjusted to 200 mM using potassium chloride, containing 0.3 mg/ml bovine serum albumin (BSA) and 0.05% Tween-20 as additives to suppress glass surface interactions. Samples were filtered through a 0.2 μm syringe filter before measurement. The sample temperature was set to 298 K using an objective heater. For each sample, three individual ACFs were recorded of 10 min measurement time each.

**NMR spectroscopy.** NMR measurements were carried out on Bruker AVANCE 600, 700, 800, 900, and 950 MHz spectrometers equipped with cryogenic triple resonance probes. Experiments were carried out at 298 K and a protein sample concentration of 300 μM. The proton chemical shifts of $^{13}$C, $^{15}$N-labeled L6-NTD were referenced to 2.2-dimethyl-2-silapentane-5-sulfonic acid (DSS). The heteronuclear $^{13}$C and $^{15}$N chemical shifts were indirectly referenced with the appropriate conversion factors[47]. All spectra were processed using Bruker TopSpin™ 2.1 or 3.2 and analyzed using the programs CARA[48] and CcpNmr Analysis 2.2[49].

Backbone resonance assignments of $^{13}$C, $^{15}$N-L6-NTD at pH 7 at a protein concentration of 300 μM were obtained from HNCO, HN(CA)CO, HNCA, and HNCACB spectra, whereas side chain assignments were obtained from (H)C(CO)NH, H(CCO)NH, HCCH-TOCSY, CBCACONH, HBHACONH spectra. All experiments were recorded with standard Bruker pulse sequences including water suppression with WATERGATE[50]. Side chains were additionally assigned with the help of $^{13}$C-NOESY-HSQC (mixing time 150 ms, aliphatic carbons) and $^{15}$N-NOESY-HSQC (mixing time 150 ms) experiments[51].

Ninety-six percent of the protein backbone could be assigned, including proline residues, whereas aliphatic side chain protons and carbons were assigned with 85% and 84%, respectively. The aromatic protons were assigned with 33%.

Partial backbone assignments of $^{13}$C, $^{15}$N-L6-NTD at pH 6 could be obtained at a protein concentration of 600 μM where the protein is mostly dimeric using an HNCA spectrum.

For the WT NTD at pH 7, the published backbone assignments of WT at pH 7.2 (BMRB Entry: 18262) could be directly transferred to our spectra, while published backbone assignments of WT at pH 5.5 (BMRB Entry 18480) were used to assign WT at pH 6.

For structure calculation of L6-NTD, peak picking and NOE assignment was performed with the ATNOS/CANDID module in UNIO[52] in combination with CYANA[53] using the 3D NOESY spectra listed above. Peak lists were reviewed manually to correct for artefacts. Distance restrains were obtained using the CYANA based automated NOE assignment and structure calculation protocol[53]. Torsion angle restraints were derived from backbone H, N, Cα, Cβ chemical shifts using TALOS+[54].

The final set of torsion angle and distance restraints was used to calculate 100 conformers with CYANA. Twenty structures with the lowest target function were submitted to a restrained energy refinement with OPALp[55] and the AMBER94 force field[56]. The structure was validated with the Protein Structure Validation Software suite 1.5[57] restricted to residues with hetNOE values > 0.6 (Supplementary Table 1).

{$^1$H},$^{15}$N-heteronuclear nuclear Overhauser effect (hetNOE) experiments for $^{15}$N-labeled WT-NTD and L6-NTD were recorded using Bruker standard pulse sequences. Experiments were run in an interleaved fashion with and without proton saturation during the recovery delay. Peak integrals were obtained using Bruker TopSpin 3.2. The {$^1$H},$^{15}$N-hetNOE data were recorded on a Bruker

600 MHz spectrometer in an interleaved manner with a $^1H$ saturation period of 5 s duration on-resonant or 10,000 Hz off-resonant for the cross-experiment and reference experiment, respectively. The relaxation delay was set to 3 s. Two consecutive ${^1H},^{15}N$-hetNOE experiments of both WT and L6-NTD were carried out and their average values are shown in Fig. 6c.

For H/D exchange experiments, WT-NTD and L6-NTD was lyophilized in an NMR tube, respectively, to remove $H_2O$. The dried protein was dissolved in an equivalent amount of $D_2O$ and immediately placed into the NMR spectrometer to record a series of $^1H,^{15}N$-HSQC spectra with 12 min intervals.

**Data analysis.** Equilibrium denaturation data were fitted using the thermodynamic model for two-state folding[11]. The spectroscopic signal $S$ was expressed as a function of the perturbation $P$[58]:

$$S(P) = \frac{\alpha_N + \beta_N \cdot P + (\alpha_D + \beta_D \cdot P) \cdot \exp\left(-\Delta G_{D-N}(P)/RT\right)}{1 + \exp\left(-\Delta G_{D-N}(P)/RT\right)} \quad (1)$$

where the $P$ was either heat ($T$, thermal denaturation) or concentration of urea ([urea], chemical denaturation), $\alpha_N$, $\beta_N$, $\alpha_D$, and $\beta_D$ characterized the linearly sloping baselines of native (N) and denatured (D) states, $R$ was the gas constant, and $\Delta G_{D-N}$ the difference in free energy between D and N.

$\Delta G_{D-N}$ as a function of denaturant is described by the linear-free energy relationship[59]:

$$\Delta G_{D-N}([\text{urea}]) = \Delta G_{D-N} - m_{D-N}[\text{urea}] \quad (2)$$

where $m_{D-N}$ is the equilibrium $m$-value that describes the sensitivity of the folding equilibrium to denaturant. Experimental errors of $\Delta G_{D-N}$ were determined from propagated errors (s.e.) of fitted values of $m_{D-N}$ and mid-point concentrations of urea ([urea]$_{50\%}$).

Analysis of thermal denaturation data to determine $T_m$ was performed using Eq. 1 in combination with the Gibbs–Helmholtz formalism[11].

Kinetic transients of folding/unfolding measured using stopped-flow spectroscopy were fitted to a single exponential function containing a linear baseline drift:

$$S(t) = a \exp(-k_{obs}t) + bt + c \quad (3)$$

$S(t)$ was the fluorescence signal as function of time, $a$ was the amplitude and $k_{obs}$ the observed rate constant. The parameters $b$ and $c$ described the linear drift of the baseline[11]. $k_{obs}$ is the sum of the microscopic rate constants for folding and unfolding ($k_f$ and $k_u$). The change of $k_{obs}$ as a function of denaturant concentration was analyzed by fitting the data to the chevron model for a barrier-limited two-state transition that follows the linear-free-energy relationship[60]:

$$\log k_{obs}([\text{urea}]) =$$
$$\log\left[k_f \exp\left(-\frac{m_{TS-D}[\text{urea}]}{RT}\right) + k_u \exp\left(\frac{m_{TS-N}[\text{urea}]}{RT}\right)\right] \quad (4)$$

$m_{TS-D}$ and $m_{TS-N}$ were the kinetic $m$-values of folding and unfolding, respectively, where TS denoted the transition state of folding. $k_f$ and $k_u$ were the microscopic rate constants of folding and unfolding, respectively, under standard solvent conditions and in the absence of denaturant.

Dimerization (binding isotherms) was analyzed using two different thermodynamic models. In thermophoresis experiments we used fluorescently modified NTD that bound non-modified NTD present at excess concentration. This pseudo-dimerization equilibrium could be described using the model for a bimolecular binding isotherm (A + B = AB). The concentration of the complex AB was described as[61]:

$$[AB] = \frac{[A]_t[B]}{[B] + K_d}, \quad (5)$$

where [AB] was the concentration of the complex (pseudo-dimer), $[A]_t$ was the total concentration of fluorescently modified NTD, [B] was the concentration of non-modified NTD and $K_d$ was the equilibrium dissociation constant. The fluorescence signal and thermophoresis ratio was modeled as:

$$\frac{S - S_u}{S_b - S_u} = \frac{[B]}{[B] + K_d}, \quad (6)$$

where $S$ was the observed signal, $S_u$ was the signal of the unbound state and $S_b$ was the signal of the bound state.

Analysis of high-resolution SEC data was carried out by plotting $V_E$ versus concentration of NTD, which yielded a binding isotherm of dimerization. The isotherm was fitted using a thermodynamic model for a monomer/dimer equilibrium ($2N = N_2$), where N and $N_2$ denoted for NTD in the monomeric and dimeric form, respectively. $K_d$ was described by the law of mass action:

$$K_d = \frac{[N]^2}{[N_2]} \quad (7)$$

Assuming reversible dimerization, measured values of $V_E$ are composed of:

$$V_E = V_E^N + F_{N2}\left(V_E^{N2} - V_E^N\right) \quad (8)$$

Where $V_E^N$ and $V_E^{N2}$ were the elution volumes of monomer and dimer, and $F_{N2}$ is the fraction of dimer, which was described as:

$$F_{N2} = \frac{[N_2]}{c_t - [N_2]} \quad (9)$$

Where $c_t$ is the total concentration of NTD in terms of monomer. $[N_2]$ was described as:

$$[N_2] = \frac{K_d}{8}\left[1 - \left[\left(1 + \frac{4c_t}{K_d}\right)^2 - \frac{16c_t^2}{K_d^2}\right]^{0.5}\right] + \frac{c_t}{2} \quad (10)$$

In PET-FCS experiments ACFs, $G(\tau)$, were fitted using an analytical model for translational diffusion of a globule that exhibited two independent, single-exponential relaxations[17]:

$$G(\tau) = \frac{1}{N\left(1 + \frac{\tau}{\tau_D}\right)}\left(1 + a_1\exp\left(-\frac{\tau}{\tau_1}\right) + a_2\exp\left(-\frac{\tau}{\tau_2}\right)\right) \quad (11)$$

$\tau$ was the lag time, $N$ was the average number of molecules in the detection focus, $\tau_D$ was the diffusion time constant, $a_1$ and $a_2$ were the amplitudes of the first and second relaxation, and $\tau_1$ and $\tau_2$ were the corresponding time constants. The application of a model for diffusion in two dimensions was of sufficient accuracy because the two horizontal dimensions ($x$, $y$) of the detection focus were much smaller than the lateral dimension ($z$) in the applied setup[17]. With $\tau_i = 1/k_i$, microscopic time constants ($\tau_{out}$ and $\tau_{in}$) were calculated from the observed amplitudes and time constants, $a_1$ and $\tau_1$[33]:

$$k_1 = k_{out} + k_{in}; a_1 = k_{out}/k_{in} \quad (12)$$

Errors are s.e. from regression analysis and propagated s.e.

### Data availability

The protein data bank (PDB) accession code for the solution NMR structure of L6-NTD is 6QJY (https://www.rcsb.org/). The data that support the findings of this study are available from the corresponding author upon reasonable request. The source data underlying Figs. 1b, 2a–e, 3, 4, 5a, b, 6, and 8 are provided as a Source Data file.

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

## Acknowledgements

We thank Carolin Hacker for stimulating discussions. The authors are grateful to the U.S. Army Research Office for financial support (grant number W911NF-17-1-0336) to H.N. B.G. acknowledges a PhD fellowship from the Max Planck Graduate Center (MPGC). U.A.H. acknowledges support by the Carl Zeiss Foundation and the Center of Biomolecular Magnetic Resonance (BMRZ), Goethe University Frankfurt, funded by the state of Hesse.

## Author contributions

J.C.H. designed experiments, synthesized protein material, performed denaturation experiments, far-UV CD, Trp fluorescence, stopped-flow fluorescence spectroscopy, PET-FCS, thermophoresis, analytical SEC, analyzed data, and created figures. B.G. performed NMR experiments, NMR structure calculation, analyzed NMR data, and created figures. C.M.J. performed SEC-MALS and thermophoresis experiments, and analyzed data. U.A.H. designed and performed NMR experiments, analyzed NMR data, and wrote the paper. H.N. conceptually designed the research, designed experiments, analyzed data, created figures, and wrote the paper.

## Competing interests

The authors declare no competing interests.
