## [Peer Review File · Nature Communications]

Reviewers' comments:

Reviewer #1 (Remarks to the Author):

The manuscript entitled "Methionine in a protein hydrophobic core drives tight interactions required for assembly of spider silk" proposes the elucidation of the roles of the methionine in the dimerization process of the N-terminal domain (NTD) of the spidroin MaSp1 of the spider *Euprosthenoops australis*. The scientific approach is clearly presented.

The authors start from the observation that the sequence of the N-terminal domain of MaSp from different species contain a high average content of methionines compared to the vertebrate and dragline silk proteins. In order to elucidate the potential importance of these methionines in the protein functionality, they compared the structure and the dynamics of the wild type NTD and mutants whose methionines were replaced by leucines. The authors present different studies: equilibrium and kinetic folding experiments, dimerization studies and the analysis of protein dynamics with a combination of various methods like CD, fluorescence, SEC-MALS, NMR, or PET-FCS... The study is elegant and clearly presented; the results obtained are convincing and show that the mutations do not influence the structure of the protein but its dynamics and its dimerization ability, suggesting the importance of methionines for the protein function. This work help to gain insight into spider silk proteins behavior.

Minor comments:

-Page 4: the authors discuss the average content of methionine in different proteins. In order to highlight this work, I suggest performing the same analysis and discussion on other amino acids in vertebrate proteins and/or in proteins with structure similar to MaSp1 NTD.

-Figure 1c (in the third, zooming view): the sequence numbers of the mutated amino acids are not easily distinguishable. Could the authors change the color of the police script?

-Page 7: in the dimerization studies using SEC-MALS: could the authors precise the molecular weights expected for the monomer and the dimer? Moreover, the authors should briefly explain why the determined dimer molecular weight of the WT protein does not correspond to the double value of the monomer molecular weight (figure 3a).

-Page 14, line 326: the sentence: "replacement of Met may thus destabilize the denaturated state through elimination of non-native interactions, ...folding" is important but not clear. Could the authors revise it?

-Figure 2c, inset; figure 3e, and figure 4b: the increasing effects of the cumulative mutations on the protein behavior seem to be reach a shoulder between the L3 and the L4 mutants. Do the authors have an explanation for that?

Reviewer #2 (Remarks to the Author):

Heiby et al describe a well-designed and -performed study with interesting results concerning a Met to Leu replaced N-terminal domain from a major ampullate spidroin (L6-NTD). The main finding is that replacing Met with Leu makes the NTD core more stably packed which in turn influences dynamic fluctuations so that pH mediated monomer to dimer transition, in the way it occurs in the wildtype protein, is precluded in L6-NTD.

The nature of the pH6 dimer remains to be characterized; the authors don't have assignments of L6-NTD at pH6, which would allow illustrating the chemical shift perturbations due to pH change on the structure and also compare them to WT-NTD. Besides, it would be interesting to compare the L6-NTD at pH6 with a mutant that in addition harbours Glu to Gln mutations that in WT-NTD results in a constitutive dimer irrespective of pH (eg L6+E79Q/E84Q/E119Q).

Another open question is to what extent the significantly more hydrophobic nature of Leu compared to

Met (see eg Nature. 2005;433:377-81) affects the results. One interesting experiment would be to study A6-NTD, since Ala, in contrast to Leu, is close to Met in terms of biological hydrophobicity (see reference above). This will also address the question whether Met has special properties that are necessary for NTD function, as suggested by Heiby et al, or if the abundance of Met rather is a coincidence reflecting the avoidance of strongly nonpolar residues.

Minor comments:

-SEC is run at pH6, what happens at pH5, which is the presumed low-end pH in the spinning duct?
-the sequence comparisons (eg Fig 6) are limited to major ampullate spidroins. To what extent are the Met conserved and overrepresented if other spidroin types are taken into consideration?

Jan Johansson

Reviewer #3 (Remarks to the Author):

The authors report a very thorough study that delivers surprising results that have potentially far-reaching implications. This manuscript is extremely well suited for publication in Nature Communications. The paper should have a high impact in the large world of protein science, with particularly strong influence on efforts to understand protein evolution and to engineer proteins with new or modified functions. This paper deserves a broad audience.

The work is motivated by a desire to understand the relationship between spider silk properties and the structure of protein components of the silk. In particular, the authors focus on the N-terminal domain of spidroin. This domain was already known to undergo pH-dependent changes in tertiary structure that lead to changes in quaternary structure (monomer vs. dimer). This change is essential for the ability of silk proteins to transform from their soluble storage form to silk fibers upon extrusion.

The experiments described here were grown out of the authors' having noticed that the hydrophobic core of the NTD contains more methionine residues than one would expect on a statistical basis. Met is quite rare, but it is also hydrophobic, so this presence of multiple buried Met residues is not necessarily surprising, but the results show that the authors were rewarded for their keen observation.

To ask whether the Met residues play any special role in terms of NTD function, they mutated all six to Leu. I think many protein scientists would have predicted that this substantial change would lead to lower tertiary structure stability, but the first surprising discovery is that the Leu-6 variant is much more stable than the WT NTD. However, the authors then show that this mutant is less functional than WT, because the mutant dimerizes much more weakly than the WT -- another surprise.

A great deal of characterization, involving many complementary analytical methods, leads to a fascinating and convincing explanation for the relationships among structure and function in the WT protein and its Leu-6 mutant. The authors make a convincing case that the Met residues confer a functionally important degree of conformational plasticity on the WT NTD, and that this plasticity is critical in its effect on features at the surface of the protein. Yet another surprise. This surprise is augmented by the evidence that incremental Met-to-Leu changes lead to incremental changes in protein flexibility and function.

Everyone engaged in protein design will want to study this example carefully, and I am sure that many laboratories will be inspired by this paper to explore the impact of replacing residues with

aliphatic side chains (Leu, Ile or Val) in the cores of their favorite proteins.

The paper is written in a logical way and very easy to follow. The experimental design is perfectly logical. My only very minor suggestion is that the authors might want to reference and perhaps provide a comment on a very recent paper that describes another methionine-rich protein (Kato et al. Cell 177:711 (2019)). I do not see an obvious connection between this recent paper and the authors' work, other than the multitude of methionine residues, but perhaps the authors will have a deeper perspective.

I look forward to being able to cite this excellent work. My group will modify its experimental designs based on the unexpected lessons that this paper teaches us.

Response to the referee comments on NCOMMS-19-14681

We thank all three referees for their time and effort reviewing our manuscript and for their valuable comments. In the following, we address the comments (cited in *italics*) point by point and highlight changes made in the revised manuscript.

Reviewer 1:

Comment:

The manuscript entitled “Methionine in a protein hydrophobic core drives tight interactions required for assembly of spider silk” proposes the elucidation of the roles of the methionine in the dimerization process of the N-terminal domain (NTD) of the spidroin MaSp1 of the spider Euprosthenops australis. The scientific approach is clearly presented.

The authors start from the observation that the sequence of the N-terminal domain of MaSp from different species contain a high average content of methionines compared to the vertebrate and dragline silk proteins. In order to elucidate the potential importance of these methionines in the protein functionality, they compared the structure and the dynamics of the wild type NTD and mutants whose methionines were replaced by leucines. The authors present different studies: equilibrium and kinetic folding experiments, dimerization studies and the analysis of protein dynamics with a combination of various methods like CD, fluorescence, SEC-MALS, NMR, or PET-FCS... The study is elegant and clearly presented; the results obtained are convincing and show that the mutations do not influence the structure of the protein but its dynamics and its dimerization ability, suggesting the importance of methionines for the protein function. This work help to gain insight into spider silk proteins behavior.

Minor comments:

-Page 4: the authors discuss the average content of methionine in different proteins. In order to highlight this work, I suggest performing the same analysis and discussion on other amino acids in vertebrate proteins and/or in proteins with structure similar to MaSp1 NTD.

Response:

We thank the reviewer for the positive assessment of our work.

We agree with the reviewer that a broader analysis of the NTD amino acid composition is very helpful, before focusing on methionine. We therefore introduced a separate sub-section at the beginning of Results that describes the unusual contents of various amino acids in the NTD in comparison to common proteins. This section can be found on page 4 of the revised manuscript.

Comment:

Figure 1c (in the third, zooming view): the sequence numbers of the mutated amino acids are not easily distinguishable. Could the authors change the color of the police script?

Response:

It was difficult to improve visibility of sequence numbering of mutated side chains by changing the colour or font. We re-assessed the figure and realized that the numbers in the molecular graphics image are actually rather confusing. We therefore deleted them, with the exception of conserved W10, which is indicated in colour. We feel that the revised figure gained clarity from

this modification.

Comment:

Page 7: in the dimerization studies using SEC-MALS: could the authors precise the molecular weights expected for the monomer and the dimer? Moreover, the authors should briefly explain why the determined dimer molecular weight of the WT protein does not correspond to the double value of the monomer molecular weight (figure 3a).

Response:

The molecular weights of the NTD monomer/dimer are 14/28 kDa. We now specified these values in the revised manuscript on page 8. The value of WT-NTD measured at pH 6.0 was 26 kDa, which is only 7% below the value expected for the dimer. This discrepancy is little above to the precision of the measurement ($\pm 3-5\%$) and may be explained by the application of small amounts of salt (60 mM ionic strength) in SEC experiments, which is required reduce sticking of protein material to the column. Salt shields electrostatics and dissociates the NTD (Refs 8, 9 and 11 in the revised manuscript), which can lead to residual population of monomer in the dimer elution band (monomer and dimer are in rapid, dynamic equilibrium), and consequently a lower detected molecular mass. We explained and discussed this observation on page 8 of the revised manuscript.

Comment:

Page 14, line 326: the sentence: “replacement of Met may thus destabilize the denaturated state through elimination of non-native interactions, ...folding” is important but not clear. Could the authors revise it?

Response:

We agree with the reviewer that the destabilizing effect of Met-to-Leu mutations on the denatured state requires more explanation. The sulphur atom in Met is proposed to make the side chain “sticky” at van der Waals contact (Ref. 36 of the revised manuscript). Following this proposal, Met side chains interact stronger with other hydrophobic groups in the denatured state of the protein, which will reduce its free energy. Mutating Met will abolish the effect and thus increase the free energy of the denatured state, which is now closer to the free energy of the transition state (Fig. 2f in the revised manuscript). The reduced difference of free energy between denatured state and transition state will increase the rate constant of folding, which is what we measured. A corresponding explanation is now given on page 16 of the revised manuscript, where we refer to Fig. 2f, which illustrates the explanation.

Comment:

Figure 2c, inset; figure 3e, and figure 4b: the increasing effects of the cumulative mutations on the protein behavior seem to be reach a shoulder between the L3 and the L4 mutants. Do the authors have an explanation for that?

Response:

The L4 construct adds mutation M48L to construct L3. This mutation appears to have little effect on stability and on dimerization of the NTD, as pointed out by the reviewer. The structure shows that the M48 side chain is positioned between helix 2 and helix 3 and thus not fully in core position. It is therefore likely that Met at position 48 experiences a less consolidated van-der-Waals interaction network than other Met side chains probed. This may explain the reduced effect of mutation M48L. On the other hand, Met is reported (Pal et al., J Biomol Struct Dyn 2001, 19, 115-128, Ref. 25 in the revised manuscript) to interact specifically with Trp side chains, which may cause a stabilizing effect. The loss of stability caused by removing this interaction may be compensated by an increase of stability upon mutation to Leu, as observed the other sites. We added a more comprehensive explanation to page 6 of the revised manuscript.

Reviewer 2:

Comment:

Heiby et al describe a well-designed and –performed study with interesting results concerning a Met to Leu replaced N-terminal domain from a major ampullate spidroin (L6-NTD). The main finding is that replacing Met with Leu makes the NTD core more stably packed which in turn influences dynamic fluctuations so that pH mediated monomer to dimer transition, in the way it occurs in the wildtype protein, is precluded in L6-NTD.

The nature of the pH6 dimer remains to be characterized; the authors don't have assignments of L6-NTD at pH6, which would allow illustrating the chemical shift perturbations due to pH change on the structure and also compare them to WT-NTD. Besides, it would be interesting to compare the L6-NTD at pH6 with a mutant that in addition harbours Glu to Gln mutations that in WT-NTD results in a constitutive dimer irrespective of pH (eg L6+E79Q/E84Q/E119Q).

Response:

We thank the reviewer for this positive assessment of our work.

We agree with the reviewer that NMR assignments of the L6-NTD dimer at pH 6 are desirable. We therefore recorded additional NMR data at pH 6.0, performed assignment of residues and analysed the chemical shift changes (new Fig. 5d and Supplementary Fig. 2 of the revised manuscript). Most residues could be assigned, but the assignment of residues of helices 2 and 3 in the dimerization interface of L6-NTD were complicated by line broadening due to the reduced dimer affinity and thus intermediate exchange processes on the NMR time scale. Importantly, this finding supports our observation that the mutation of Met to Leu residues in the protein core has profound consequences both for protein dynamics and function and prevents the dimer interface to adopt a conformation suitable for high-affinity dimerization. Indeed, our high-resolution SEC experiments showed that the L6-NTD dimer is loosely associated with dimensions larger than those of the wild-type dimer (discussed on pages 9 and 15 of the revised manuscript). Loose association is presumably associated with rapid inter-molecular interactions dynamics between subunits, which may enhance NMR line broadening. In addition, NMR chemical shift differences between pH 6 and pH 7 are significantly more

pronounced in the WT-NTD compared to L6-NTD. This observation substantiates the finding of strong dimer interactions and conformational change in WT-NTD, which are missing in L6-NTD. We described these additional results on page 10-11 and showed them as new panel (d) in Fig. 5 of the revised manuscript.

The introduction of triple mutant E79Q/E84Q/E119Q, which generates a constitutive dimer in WT-NTD irrespective of pH (Ref. 12 in the revised manuscript), on the background of L6-NTD is a very interesting experiment suggested by the reviewer. It addresses dissection of the multi-step dimerization of the NTD reported in Ref. 12 and confirmed here. It may further dissect electrostatic contributions in dimerization from conformational change blocked by the L6 mutations. But given the already comprehensive data set presented in this manuscript we believe an additional triple-mutant experiment on L6-NTD is beyond the scope of this work but will be performed within the scope of a follow-up study.

Comment:

Another open question is to what extent the significantly more hydrophobic nature of Leu compared to Met (see eg Nature. 2005;433:377-81) affects the results. One interesting experiment would be to study A6-NTD, since Ala, in contrast to Leu, is close to Met in terms of biological hydrophobicity (see reference above). This will also address the question whether Met has special properties that are necessary for NTD function, as suggested by Heiby et al, or if the abundance of Met rather is a coincidence reflecting the avoidance of strongly nonpolar residues.

Response:

There are significant differences among hydrophobicity scales of amino acids in the literature (Rose et al. Science 1985, 229, 834-838; Ref. 21 in the revised manuscript). The reviewer refers to a paper (Nature. 2005;433:377-81) that uses insertion of helices into a cellular membrane as a probe for hydrophobicity. This situation is different from burial of hydrophobic side chains in a protein core investigated here. Hydrophobicity scales in the protein folding literature show that Met and Leu have very similar hydrophobicity but deviate significantly from Ala (Chothia, Annu Rev Biochem 1984, 53, 537-572; Rose et al. Science 229, 4716, 83-838; Refs. 20 and 21 in the revised manuscript). This is also reflected in their Nozaki-Tanford free energies of transfer from water to organic solvent (Ref. 21). Moreover, the residue side chain volumes of Met and Leu are virtually identical while that of Ala is substantially smaller (Ref. 20). We refer to refs. 20 and 21 in the Results section on page 5 of the revised manuscript.

Regarding the reviewer's second point of introducing Ala instead of Met, we previously introduced a single-point mutation replacing core Met77 with Ala (mutant M77A, unpublished results). Unfortunately, the expression of this mutant failed, which may be explained by a strongly destabilizing effect of a so-called deletion mutation: M77A replaces a large thioether side chain by the small methyl group, which deletes a substantial number of van-der-Waals contacts in the domain core. M77A thus effectively generates a small hole in the core. Therefore, based on our experience with mutant M77A, an A6 equivalent of the L6-NTD most likely cannot be prepared.

Comment:

Minor comments:

-SEC is run at pH6, what happens at pH5, which is the presumed low-end pH in the spinning duct?

Response:

We performed dimerization experiments at pH 6.0 because in many previous biophysical studies this has been a reference pH for the dimerized state and the change in Trp fluorescence associated with dimerization of NTDs reaches an end point at pH 6 (e.g. Ref. 10 of the revised manuscript). A pH of 5.7 has been measured in the distal part of a spinning duct (Ref. 27 in the revised manuscript), which is close to pH 6.0. We agree with the reviewer that measurements at pH 5 are of interest because this may be the presumed end-point in the very distal part of the spinning duct close to the exit spigot. Following the suggestion of the reviewer, we performed SEC runs of the L6-NTD in pH 5.2 buffer at various protein concentrations (Figure below). The result shows that the L6-NTD dimer is further stabilized at pH 5, which is evident from the later onset of dissociation upon dilution ($K_d < 1 \mu\text{M}$). The observation can be explained by further protonation of L6-NTD and enhanced electrostatic interactions between the subunits. However, under these conditions the WT-NTD dimer was still substantially more stable than L6-NTD because WT-NTD did not show any onset of dissociation in the measured concentration range (Figure below). We were unfortunately not able to determine a reliable K_d of L6-NTD under these conditions. The reason for this is that sample concentrations below 50 nM failed to generate a measurable Trp fluorescence signal at the detector. The proton concentration at pH 5 is ten times higher than at pH 6 and a consequently more heavily protonated NTD sample presumably tends to stick stronger to the negatively charged silica glass wall surface of the flow cell in the detection unit of the HPLC instrument, which may explain the loss of signal. Given these complications and uncertainties in the interpretation of data, we chose not to show and discuss the pH 5 data in the revised manuscript, but instead show and discuss them here within the scope of a response to the reviewer comment.

Figure: SEC of L6-NTD and WT-NTD at pH 5.2. Elution volumes (V_E) of L6-NTD (orange) and WT-NTD (blue) measured as a function of protein concentrations in 20 mM MES pH 5.2 with the ionic strength adjusted to 20 mM using potassium chloride.

Comment:

-the sequence comparisons (eg Fig 6) are limited to major ampullate spidroins. To what extent are the Met conserved and overrepresented if other spidroin types are taken into consideration?

Response:

We thank the reviewer for the good idea to analyse overrepresentation and conservation of Met residues also in NTDs from other spidroin types. We therefore revised the analysis and included additional sequences of spidroin NTDs from the minor ampullate, tubuliform, cylindrical and aciniform glands described in ref. 15 of the revised manuscript (new Figures 7 and 8). The analysis interestingly shows that Met is similarly overrepresented in NTDs from major ampullate, minor ampullate and aciniform spidroins. In contrast, Met is missing in NTDs from tubuliform and cylindrical spidroins altogether. It is tempting to correlate the result with mechanical properties of the different silk types, where silks containing Met-rich high-affinity spidroin NTDs should exhibit higher strength compared to the ones containing Met-depleted low-affinity spidroin NTDs. Major and minor ampullate, as well as aciniform silks are used to build the orb web and for prey wrapping and exhibit high toughness (Ref. 44 in the revised manuscript). The tubuliform silk is a flocculent silk used to build the egg case, same as cylindrical silk. Tubuliform silk has a low toughness compared to major ampullate and minor ampullate and aciniform silk (Ref. 44 in the revised manuscript). The correlation of the content of Met in NTDs with silk toughness underscores its important role in enhancing the strength of spidroin connectivity. We added a corresponding discussion on page 16-17 of the revised manuscript.

Reviewer 3:

Comment:

The authors report a very thorough study that delivers surprising results that have potentially far-reaching implications. This manuscript is extremely well suited for publication in Nature Communications. The paper should have a high impact in the large world of protein science, with particularly strong influence on efforts to understand protein evolution and to engineer proteins with new or modified functions. This paper deserves a broad audience.

The work is motivated by a desire to understand the relationship between spider silk properties and the structure of protein components of the silk. In particular, the authors focus on the N-terminal domain of spidroin. This domain was already known to undergo pH-dependent changes in tertiary structure that lead to changes in quaternary structure (monomer vs. dimer). This change is essential for the ability of silk proteins to transform from their soluble storage form to silk fibers upon extrusion.

The experiments described here were grown out of the authors' having noticed that the hydrophobic core of the NTD contains more methionine residues than one would expect on a statistical basis. Met is quite rare, but it is also hydrophobic, so this presence of multiple buried Met residues is not necessarily surprising, but the results show that the authors were rewarded for their keen observation.

To ask whether the Met residues play any special role in terms of NTD function, they mutated all six to Leu. I think many protein scientists would have predicted that this substantial change would lead to lower tertiary structure stability, but the first surprising discovery is that the Leu-6 variant is much more stable than the WT NTD. However, the authors then show that this mutant is less functional than WT, because the mutant dimerizes much more weakly than the WT -- another surprise.

A great deal of characterization, involving many complementary analytical methods, leads to a fascinating and convincing explanation for the relationships among structure and function in the WT protein and its Leu-6 mutant. The authors make a convincing case that the Met residues confer a functionally important degree of conformational plasticity on the WT NTD, and that this plasticity is critical in its effect on features at the surface of the protein. Yet another surprise. This surprise is augmented by the evidence that incremental Met-to-Leu changes lead to incremental changes in protein flexibility and function.

Everyone engaged in protein design will want to study this example carefully, and I am sure that many laboratories will be inspired by this paper to explore the impact of replacing residues with aliphatic side chains (Leu, Ile or Val) in the cores of their favorite proteins.

The paper is written in a logical way and very easy to follow. The experimental design is perfectly logical. My only very minor suggestion is that the authors might want to reference and perhaps provide a comment on a very recent paper that describes another methionine-rich protein (Kato et al. Cell 177:711 (2019)). I do not see an obvious connection between this recent paper and the authors' work, other than the multitude of methionine residues, but perhaps the authors will have a deeper perspective.

I look forward to being able to cite this excellent work. My group will modify its experimental designs based on the unexpected lessons that this paper teaches us.

Response:

We thank the reviewer for this positive and enthusiastic assessment of our work, and for his/her summary of the novel and surprising key points of the study. We were inspired by this summary and borrowed it to further improve our Discussion section on page 14 of the revised manuscript.

We also thank the reviewer for bringing the paper by Kato et al. to our attention. It reports on a Met-rich domain of yeast ataxin-2 that uses reversible oxidation of solvent-exposed Met to facilitate phase transition of the protein into gel-like states. The phenomenon is based on the high oxidation potential of the thioether side chain of Met and the change of polarity of a sulfoxide compared to a thioether. Oxidation of sulphur in solvent-exposed Met presumably also stiffens the side chain and thus has an effect on dynamics. We included and discussed this reference on page 15 (Ref. 39) of the revised manuscript.

REVIEWERS' COMMENTS:

Reviewer #2 (Remarks to the Author):

The authors have done a good job in responding to all questions and have improved the manuscript even further. I have no further comments to offer.

Reviewer #3 (Remarks to the Author):

I carefully reviewed the authors' response to comments from Reviewer 1, as detailed in the authors' cover letter. It appears that the authors have considered each point very carefully, and responded to each with appropriate changes or additions to the manuscript.

I had only one very small suggestion for the authors when I reviewed the original manuscript, and they responded appropriately. Therefore, I recommend that this manuscript be accepted in the present form.